# Tooth-Cutting-Induced Maxillary Malocclusion Exacerbates Cognitive Deficit in a Mouse Model of Vascular Dementia

**DOI:** 10.3390/brainsci13050781

**Published:** 2023-05-10

**Authors:** Young-Jun Lee, Chiyeon Lim, Sehyun Lim, Suin Cho

**Affiliations:** 1Lee Young Jun Clinic of Korean Medicine, Institute of TMJ Integrative Medicine, Cheonan 31141, Republic of Korea; yjleejun@naver.com; 2Department of Radiology, Massachusetts General Hospital and Harvard Medical School, Boston, MA 02129, USA; rachun@hanmail.net (C.L.); favor119@daum.net (S.L.); 3College of Medicine, Dongguk University, Goyang 10326, Republic of Korea; 4School of Public Health, Far East University, Eumseong 27601, Republic of Korea; 5School of Korean Medicine, Pusan National University, Yangsan 50612, Republic of Korea

**Keywords:** temporomandibular joint, complementary medicine, vascular cognitive impairment, vascular dementia

## Abstract

Treatments to restore the balance of the temporomandibular joint (TMJ) are performed in the field of complementary and alternative medicine; however, evidence supporting this approach remains weak. Therefore, this study attempted to establish such evidence. Bilateral common carotid artery stenosis (BCAS) operation, which is commonly used for the establishment of a mouse model of vascular dementia, was performed, followed by tooth cutting (TEX) for maxillary malocclusion to promote the imbalance of the TMJ. Behavioural changes, changes in nerve cells and changes in gene expression were assessed in these mice. The TEX-induced imbalance of the TMJ caused a more severe cognitive deficit in mice with BCAS, as indicated by behavioural changes in the Y-maze test and novel object recognition test. Moreover, inflammatory responses were induced via astrocyte activation in the hippocampal region of the brain, and the proteins involved in inflammatory responses were found to be involved in these changes. These results indirectly show that therapies that restore the balance of the TMJ can be effectively used for the management of cognitive-deficit-related brain diseases associated with inflammation.

## 1. Introduction

The temporomandibular joint (TMJ), which is one of the most complex joints in the human body, mainly plays a role in the occlusion of teeth, speaking, chewing and swallowing [1,2]. The impairment of these functions is termed temporomandibular joint disorder (TMD) [1,3]. TMD can also be triggered by structural problems caused by tension in the joints that occur during chewing, swallowing, or talking; thus, TMD represents a complex health problem and requires proper management [4,5]. In a recently published study, the prevalence of TMD was 31% in adults and 11% in children and adolescents [6].

Recent studies reported that TMD causes neuropathological changes and can be accompanied by various problems caused by pain in the TMJ [7,8,9]. However, studies on the effect of TMD on pathological changes in pre-existing diseases remain insufficient. Therefore, in this study, we attempted to confirm the effects of tooth structure imbalance in a rodent model of vascular dementia (VD).

As the average life expectancy increases and the proportion of the elderly population increases rapidly, the incidence of dementia is also gradually increasing. Three out of four patients with dementia suffer from severe mental and economic burdens, such as depression or reduced life satisfaction, which affect not only the patients themselves but also their families. These factors impose a huge mental and economic burden on individuals with dementia, and the social and national costs required for managing these patients are also quite high [10,11].

Cognitive memory disorders, such as Alzheimer’s disease (AD) and VD, can be caused by various factors, with the best-known cause being clearly related to ageing. However, the results of a series of studies suggest that changes in oxygen transport to the prefrontal cortex caused by missing teeth, improper prosthesis, or reduced occlusal force are very likely to be risk factors for cognitive impairment [12,13,14,15,16].

Since there is no effective treatment for degenerative changes in the TMJ, joint replacement surgery is the only option. These degenerative changes become more severe with age; however, the pathological mechanisms related to them have not been clearly identified [17]. As the average human lifespan continues to increase because of the development of medical technology and hygiene management, the number of patients suffering from degenerative diseases of the TMJ is expected to increase, and other diseases caused by TMJ imbalance are expected to become more diverse and severe. Therefore, the purpose of this study was to identify the changes in cognitive function that occur after inducing damage to the teeth in mice with VD; in addition, we aimed to provide a basis for the method of correction of maxillary malocclusion, for its effective use to improve cognitive function.

## 2. Materials and Methods

### 2.1. Animals

Six-week-old male C57BL/6 mice (weight, 20–22 g) were purchased from Samtako Bio (Osan, Korea). The animals were housed in temperature- and humidity-controlled polypropylene cages under a light/dark cycle (12–12 h) at 24 ± 4 °C and fed standard pellet diets with free access to water for at least 1 week prior to experimentation. All experiments were approved by the Animal Care Ethics Committee of Pusan National University (approval number: PNU 2020-2540) and certified by the Korea Laboratory Animal Association.

### 2.2. Reagents

Saline was obtained from JW Pharmaceutical Co., Ltd. (Seoul, Korea); phosphate-buffered saline (PBS) was purchased from Bio Basic Inc. (Markham, ON, Canada) and optimal cutting temperature (OCT) compound cryostat embedding medium was purchased from Thermo Fisher Scientific (Waltham, MA, USA). The BCA reagent and bovine serum albumin (BSA) standards were purchased from Thermo Fisher Scientific (Waltham, MA, USA). Paraformaldehyde (PFA) was purchased from Thermo Fisher Scientific (Waltham, MA, USA); povidone-iodine was obtained from Green Pharmaceuticals Co. (Seoul, Korea) and was used for skin disinfection before and after surgical operation.

### 2.3. Preparation of the VD Mouse Model

The VD mouse model was induced using the bi-common carotid artery stenosis (BCAS) procedure (Figure 1). Briefly, the mice were anaesthetised using isoflurane (JW Pharmaceutical Co. Seoul, Korea) in N_2_O/O_2_ (70%/30%) gas until they showed no response to mechanical stimulation of the tail. A cervical midline incision was performed and the isolated common carotid artery (CCA) was wrapped with a micro-coil (inner diameter, 0.18 mm; Sawane Spring Co., Hamamatsu, Japan) to induce CCA stenosis. The body temperature of the mice was maintained at 37 ± 0.5 °C using a heating pad (Harvard Apparatus^TM^, Holliston, MA, USA).

### 2.4. Induction of Maxillary Malocclusion

Maxillary malocclusion was induced by cutting a tooth (TEX). In the case of mice, the teeth were restored to their original length after about 5 days; thus, TEX was performed twice a week (Figure 1).

### 2.5. Measurement of Body Weight and Physiological Parameters

Mice were weighed weekly during the experiment to determine if reduced blood flow affected body weight, and at the end of the experiment, they were bled under deep anaesthesia using isofluorane. Blood samples were centrifuged at 1500× *g* for 15 min at 4 °C to obtain serum. The serum concentrations of electrolytes, such as sodium (Na^+^), potassium (K^+^) and chloride (Cl^−^), were measured using an electrolyte analyser (Dri-Chem 3500i, Fuji, Japan) to monitor and rule out a potential electrolyte imbalance.

### 2.6. Test for Voluntary Alternating Behaviour

The rate of voluntary alternation increases when subjects try to explore a novel object. Working memory function and exploratory behaviour were assessed using the voluntary alternation test in a Y-shaped maze (Y-maze) with three identical arms (35 cm long × 7 cm wide × 40 cm high per arm) that were positioned at the same angle (Figure 2A). Each mouse was placed in the centre of the Y-maze and allowed to explore freely for 8 min after a 2 min habituation phase on the 2 arms of the maze. In each test, voluntary maze arm entry alternations were visually recorded. Arm entry was scored when the mouse placed its 4 paws inside the arm. Spontaneous alternation was determined upon entry into three arms from consecutive selections of three sets (e.g., C–A–B, B–C–A and A–B–C). Voluntary alternation behaviour was calculated using the following equation: Percent alternation = ([Number of alternations]/[Number of total arm entries − 2]) × 100.

### 2.7. Novel Object Recognition Test (NORT) Evaluation

This test is used to assess a rodent’s affinity for a novel object compared with a familiar object [18,19]. In the adaptation phase, each mouse was allowed to explore an open field arena (grey box, 40 cm long × 40 cm wide × 40 cm high) for 5 min without objects (Figure 2B). In the first trial, two identical objects (familiar, F) were placed at opposite corners of the test arena and the experimental mouse was allowed to explore both objects for 10 min, during which time the mouse exploration was scored. In the second trial, after 20 min, mice were placed back in the arena, and one of the identical objects provided in the first trial was replaced with a new object (new, N). The time spent exploring each object at a distance of less than 2 cm was recorded for 10 min. Object search time and discriminant rate analyses were performed using the following formula: total N time/(N time + F time) for each experimental group. The two identical objects and arenas were cleaned between each trial using 70% ethanol.

### 2.8. Sacrifice and Cardiac Perfusion for Brain Harvesting

The abdomen of each mouse was cut and cardiac perfusion was performed using PBS. Briefly, the pulmonary artery was occluded, the left ventricle was pierced with a 21-gauge needle, and the needle was anchored to the ascending aorta. Immediately after the initiation of perfusion, the right atrium was cut with scissors. PBS was used for cardiac perfusion, to obtain brain tissue for RNA sequencing; moreover, the brain tissue was stored in a freezer at −80 °C prior to use. PBS and 4% PFA in PBS were used for fixation. For post-fixation, the brains were immersed in 4% PFA at 4 °C for 12 h.

### 2.9. Cryosectioning of Mouse Brains

The brains of the mice were sequentially placed in 10%, 20% and 30% sucrose solutions, frozen in OCT compound and stored in a refrigerator at −80 °C. Subsequently, 25-μm-thick brain sections were obtained using a cryostat (Leica, Wetzlar, Germany). The sections were placed on glass slides for 12 h and stored in a refrigerator at −80 °C until further use.

### 2.10. Immunofluorescence (IF) Staining

The sections were dried in a slide warmer (Convision Co., Seoul, Korea), incubated with blocking buffer (5% BSA) for 1 h at 25 °C, and then washed with blocking buffer. Diluted primary antibodies against the neuronal nuclear antigen (NeuN) and glial fibrillary acidic protein (GFAP) (catalogue numbers 94403 and 12389, respectively; Cell Signaling Technology, Danvers, MA, USA) and the tumour necrosis factor-alpha (TNF-α) (ab1793, Abcam, Cambridge, UK) were incubated with the sections overnight at 4 °C. The primary antibody was then washed off 3 times for 5 min with PBS, and the diluted secondary antibody (goat anti-mouse or goat anti-rabbit IgG H&L; Abcam, Cambridge, UK) was dropped onto the sections and incubated at 25 °C for 2 h; the sections were then washed 3 times with PBS for 5 min each. After mounting using DAPI (ab104139, Abcam, Cambridge, UK), the sections were mounted with cover slips, with the edges sealed with nail polish and observed under a fluorescence microscope (Ni-U, Nikon, Tokyo, Japan). The samples were stored in a 4 °C refrigerator for long-term use. After staining with antibodies, the colour intensity reacting to fluorescence was regarded as the protein expression level.

### 2.11. RNA Sequencing Analysis

Total RNA was isolated using the TRIzol reagent (15596058, Invitrogen^TM^, Waltham, MA, USA), and the purity and ratio of the RNA were evaluated to confirm its suitability for this study. mRNA was extracted from the total RNA, cDNA was synthesised and mass sequencing was performed. The expression level of the sham group was set to 1 and the RNA expression level in the experimental samples was obtained by comparing the expression level of the BCAS group and of the BCAS combined with the TEX group with that of the sham group. The comparison of the relative expression level was visualised as a heat map using the MeV programme (https://mev.tm4.org, accessed on 13 December 2022). Furthermore, after selecting the protein to be used for analysis, the STRING database (https://string-db.org, accessed on 23 December 2022) was used to analyse the protein–protein interaction (PPI) network. Visualisation was performed using Cytoscape (version 3.8.2), which is a software that is used for network analysis.

### 2.12. Statistical Analysis

One-way ANOVA followed by the Holm–Sidak test was performed to determine the statistical significance of the differences between groups using the SigmaPlot 12.0 programme (Systat Software Inc., Palo Alto, CA, USA). Data are expressed as the mean ± standard deviation (SD), and statistical significance was set at *p* < 0.05.

## 3. Results

### 3.1. Effects of BCAS and TEX Operation on Body Weight Changes and Physiological Parameters in Mice

No significant differences were observed among the groups regarding body weight (Figure 3A) and physiological parameters (Figure 3B) after BCAS and TEX operation. These results indicated that 4 weeks of BCAS and TEX did not yield serious side effects that may have affected the study.

### 3.2. Behavioural Changes in the Y-Maze and NORT

BCAS and TEX for 4 weeks caused a significant change in the frequency of sequential entry of mice into the 3 arms of the Y-maze, with the extent of the change being more pronounced in the TEX group (Figure 4A). In addition, the total number of times that the animals entered the three arms remained unchanged in the BCAS group but was significantly decreased in the TEX group (Figure 4B), which implies that voluntary behaviour was reduced by TEX. Similar changes were detected in NORT; however, although the time to enter zone 3 decreased in both the BCAS and TEX groups (Figure 4C), a statistically significant decrease was observed only in the TEX group compared with zone 2 (Figure 4D). These results suggest that TEX changed the behaviour of mice that preferred to explore new objects more instinctively. Although not shown in this study, in mice with only TEX-induced conditions, no significant difference was found when compared with the sham group using the Y-maze and NORT (Appendix A).

### 3.3. Effects of BCAS and TEX on Astrocyte Activation in the Hippocampal Region

NeuN is a neuronal nuclear antigen that is commonly used as a neuronal biomarker, whereas GFAP is a well-known marker of astrocyte activation after central nervous system (CNS) injury or stress [20,21,22]. The IF staining of neuronal cells and astrocytes confirmed that astrocytes were activated by BCAS operation alone and that the degree of activation increased remarkably when the operation was combined with TEX (Figure 5). In mice with only TEX-induced conditions, activation of astrocytes was not observed as in the sham group (Appendix A). Therefore, TEX application alone does not induce an inflammatory response around the hippocampus or cause a decline in cognitive function.

### 3.4. RNA Sequencing and PPI Network Analysis

The degree of gene expression was confirmed using an RNA sequencing analysis and the amount of expression changes is depicted in Figure 6A. To confirm the changes in gene expression triggered by TEX operation in BCAS-induced mice compared with the sham group, genes that showed a >2-fold increase or a 1/2-fold decrease in the BCAS group and a >3-fold increase or a 1/3-fold decrease in the TEX group were screened. A total of 490 genes were identified, of which 407 genes were upregulated and 83 genes were downregulated in the TEX group (Figure 6B). By identifying the target proteins of these genes and analysing the PPI between them, proteins that were expected to play a major role were identified; proteins such as Cd4, Aurkb, Bub1b, Apob and Apoa4 were predicted to play a central role in this process (Figure 6C). These genes encode proteins that are highly likely to undergo changes after TEX and are expected to play a major role in the occurrence and exacerbation of the symptoms caused by TEX.

## 4. Discussion

Cerebrovascular disorders (CVDs) reduce blood flow to the brain, resulting in hypoxia and inflammatory reactions. In turn, inflammatory reactions develop toward increased oxidative stress in the brain, which is accompanied by damage to the white matter and hippocampus. These pathophysiological findings ultimately manifest as symptoms of VD [23,24]. The mortality rate of VD is higher than that of AD, which is thought to be an effect of coronary artery disease, and the changes in cognitive function in patients with VD are much more variable than those observed in other diseases, such as AD. This is because vascular lesions can occur in any part of the brain where blood vessels are distributed and depend on the neural substrates that are affected [25,26].

Due to the progress in solidification along with the increase in the population, the number of people suffering from various types of dementia is increasing, and dementia is considered a major cause of various disabilities in the older population. Therefore, maintaining a healthy brain is one of the main factors that can reduce the social and economic burdens in an aging society. Among the types of dementia, the prevalence of VD is higher in Asia than in other regions, and in particular, in patients with cerebral infarction, the risk increases more than twofold [27,28].

TMD is one of the most common diseases in older people and is accompanied by headaches, migraines and noises when opening the mouth wide. Aging is one of the most representative causes of TMD, and it reduces the quality of life because of the above symptoms [17,29,30]. Therefore, since VD and TMD can appear together as aging progresses, studying how the two diseases interact with each other is necessary. A recent study reported that inflammation in the TMJ causes abnormalities in the cerebral limbic system [31]. Moreover, a treatment to control the pain caused by TMJ imbalance by adjusting the range of motion of the TMJ by attaching an implant in the ear has also been described [32]. Therefore, although evidence is scarce, the symptoms of various diseases can reportedly be improved by controlling the imbalance of the TMJ.

Recently, we attempted to determine whether the symptoms of the disease could be improved by restoring the balance of the TMJ; however, because there is insufficient evidence to date in support of its effectiveness, in this study, we aimed to indirectly confirm the contention that restoration of TMJ balance can be effectively used to improve disease. A VD mouse model was selected as a pre-clinical study model of cognitive function deficit, and it was confirmed that the cognitive function deficit became more severe when maxillary malocclusion was induced by TEX (Figure 4).

Astrocytes are characteristic star-shaped glial cells in the brain and spinal cord, and play various key roles in supporting, guiding, nurturing and signalling neural structures and activities [33,34]. GFAP is mainly expressed in the astrocytes of the CNS and is relatively frequently used in the study of various diseases, such as traumatic brain injury, stroke and brain tumour [20,35]. In our study, astrocytes were activated by BCAS and the degree of activation was greater after the administration of TEX (Figure 5). These findings indicate that astrocytes can be more easily converted to pathological conditions by TEX. The results of this study imply that the symptoms of lesions that can appear in the CVD can be worsened by TEX, thus indirectly demonstrating that the restoration of the balance of the TMJ can slow down the progression of CVD in the elderly.

In addition, no change in cognitive function or activation of astrocytes was observed in the TEX-only group (Appendix A). However, the simultaneous application of TEX and BCAS amplified pathological changes, suggesting that the group with both diseases should be managed closely, and treatment measures must be established for this group.

In the PPI network, in which gene expression was more severely affected after TEX, the most interacting protein was Cd4. Cd4 is a marker of T cells and natural killer cells and is one of the molecules involved in the immune response [36]. The application of BCAS and TEX operation mainly involves the inflammatory response; therefore, it can be assumed that the restoration of the balance of the TMJ can suppress the symptoms of various diseases by acting on the immune system and controlling the inflammatory response.

In summary, this study is the first to show that the restoration of the TMJ balance can be effectively applied to the improvement of CVD. Moreover, it has been confirmed that TEX can exacerbate BCAS-induced cognitive deficits; therefore, it is expected that complementary therapies that restore the balance of the TMJ can be effectively used for the management of various diseases accompanied by an inflammatory response.

## Figures and Tables

**Figure 1 brainsci-13-00781-f001:**
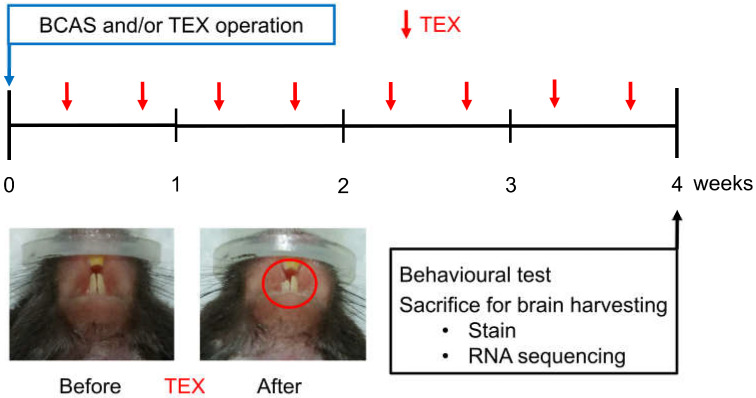
Overview of the study process aimed at observing the changes in cognitive function after inducing TEX in mice with BCAS operation. TEX was applied to mice twice a week and, after four weeks, the animals were sacrificed after measuring behavioural changes; moreover, brain tissue was obtained and used for immunofluorescence staining and RNA sequencing.

**Figure 2 brainsci-13-00781-f002:**
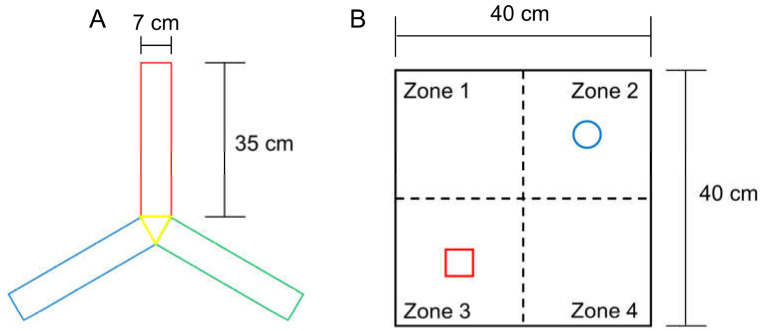
Specifications of the Y-maze (**A**) for the triplet arm alternation test, and the open field box (**B**) for NORT measurement. The size of each arm in the Y-maze was the same, and the objects in zones 2 and 3 were secured using double-sided tape to prevent them from moving easily.

**Figure 3 brainsci-13-00781-f003:**
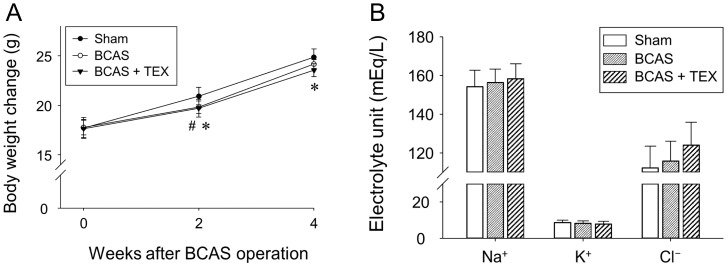
Effects of BCAS-induced VD and TEX operation on body weight changes (**A**) and physiological parameters (**B**) in mice. Mice were weighed every 2 weeks during the experimental period. Serum samples were obtained to measure the concentrations of Na^+^, K^+^ and Cl^−^. All data are expressed as the mean ± SD (n = 5). ^#^, statistically significant when the BCAS group was compared with the sham group (*p* < 0.05); *, statistically significant when the TEX group was compared with the sham group (*p* < 0.05).

**Figure 4 brainsci-13-00781-f004:**
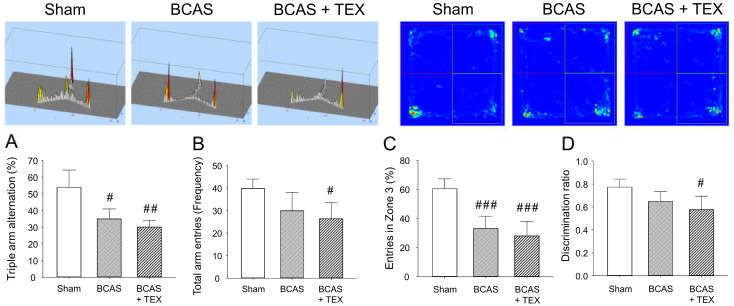
Y-maze test and NORT were used to analyse the changes in the behaviour of mice caused by BCAS and TEX-induced cognitive function deficit. Triple arm alternation (**A**) and total arm entries (**B**) were measured using the Y-maze. Entries in a specific zone (**C**) and discrimination of the novel object (**D**) were measured using the NORT. All data are expressed as the mean ± SD (n = 5). ^#^, ^##^, ^###^, statistically significant compared with the sham group (*p* < 0.05, *p* < 0.01 and *p* < 0.001, respectively).

**Figure 5 brainsci-13-00781-f005:**
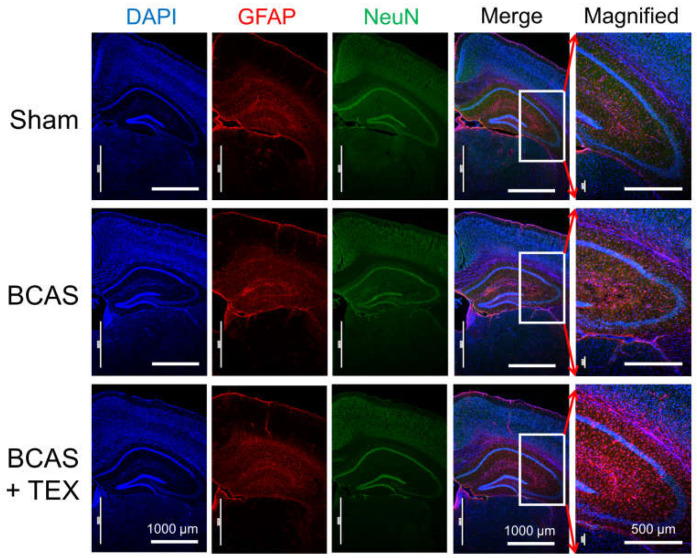
Neuroinflammatory responses in the hippocampal region of BCAS- and TEX-induced mouse brains. BCAS causes GFAP-positive astrocyte activation and TEX exacerbates the activation. Each photomicrograph (×40 or ×100) represents a region of the hippocampus that was stained using immunofluorescence.

**Figure 6 brainsci-13-00781-f006:**
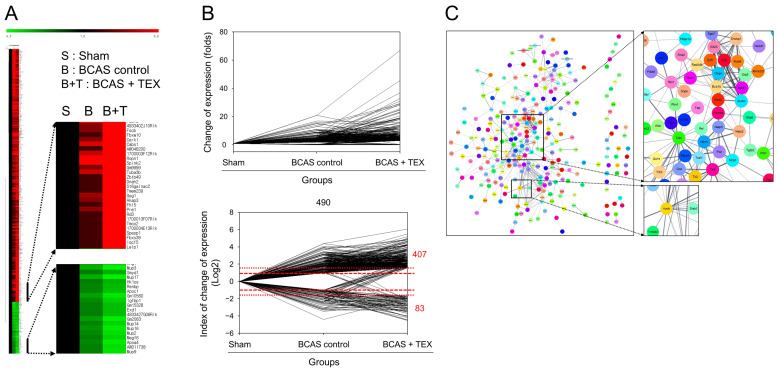
Clustering analysis of expression profiling (**A**), visualisation of gene expression changes (**B**) and PPI network construction (**C**). (**A**) Red, green and black indicate upregulation, downregulation and no change in gene expression, respectively. The colour intensity is related to the expression level of a gene. (**B**) Data were converted to log2 to easily identify the increase and decrease in the expression of genes. (**C**) Each node represents a protein and the line connecting the nodes represents an interaction. The thickness of the line indicates the strength of the interaction.

## Data Availability

The data that support the findings of this study are available from the corresponding author upon reasonable request. Further inquiries can be directed to the corresponding author.

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
