# Peer review of "Tooth-Cutting-Induced Maxillary Malocclusion Exacerbates Cognitive Deficit in a Mouse Model of Vascular Dementia"

_brainsci, 2023, doi:10.3390/brainsci13050781_

Round 1

Reviewer 1 Report

This is a very interesting original research paper regarding the role of temporo-mandibular joint (TMJ) dysfunction role in aggravating cognitive deficits.

Structural changes in the oral cavity may provoke changes in oxygen transport to the prefrontal cortex, aggravating cognitive deficits.

The authors used the rodent model of vascular dementia with bi-lateral common carotid artery stenosis (BCAS) operation. To this mice also they performed tooth-cutting for inducing maxillary malocclusion (TEX) resulting in TMJ imbalance.

They asses changes in behavior of mice using the voluntary alternating behavior test in Y-shaped maze and novel object recognition test - to BCAS and to BCAS with TEX. The authors confirmed that cognitive function deficit became more severe when maxillary malocclusion was induced by TEX.

Using immunofluorescence (IF) staining of the brain glial fibrillary acidic protein (GFAP) the authors revealed astrocyte activation in hippocampal region after BACS, even an increased activation after TEX.

Indirectly, the authors demonstrated that first to show that the restoration of the TMJ balance could improve cognitive deficits.

Author Response

This is a very interesting original research paper regarding the role of temporo-mandibular joint (TMJ) dysfunction role in aggravating cognitive deficits.

Structural changes in the oral cavity may provoke changes in oxygen transport to the prefrontal cortex, aggravating cognitive deficits.

The authors used the rodent model of vascular dementia with bi-lateral common carotid artery stenosis (BCAS) operation. To this mice also they performed tooth-cutting for inducing maxillary malocclusion (TEX) resulting in TMJ imbalance.

They asses changes in behavior of mice using the voluntary alternating behavior test in Y-shaped maze and novel object recognition test - to BCAS and to BCAS with TEX. The authors confirmed that cognitive function deficit became more severe when maxillary malocclusion was induced by TEX.

Using immunofluorescence (IF) staining of the brain glial fibrillary acidic protein (GFAP) the authors revealed astrocyte activation in hippocampal region after BACS, even an increased activation after TEX.

Indirectly, the authors demonstrated that first to show that the restoration of the TMJ balance could improve cognitive deficits.

Reply: Thank you very much for your positive evaluation.

Reviewer 2 Report

This interesting work have some doubts for me:

1- Why Do de authors not included the group of only "TEX"?

2- the inflammatory response has been evaluated using GFAP, which is an astrocyte marker. I think it would be more convenient to use a microglia marker such as IBA1 to assess the inflammatory process more efficiently and truthfully.

3-dental breakage used in the methodology (TEX) implies a greater risk of oral infections for animals. Have the authors excluded this possibility by evaluating any marker of infection?

4-Only 9 references of the 31 used are after 2018. I think the authors should carry out a deeper and more updated bibliographic review.

5-The article presents a large number of results of valuable interest, however the discussion of these results is superficial, and requires further work and deepening in which all the results obtained are framed.

Author Response

This interesting work have some doubts for me:

1- Why Do de authors not included the group of only "TEX"?

Reply: As per your opinion, we (the authors) conducted an experiment by separately classifying the experimental group that only received TEX treatment; as a result, no significant difference was found when compared with the sham operation group, so it was excluded from the manuscript. However, following authors’ discussion of your comments, it would be better to include those results in the revised manuscript. Accordingly, we presented them as a supplementary material. Thank you for your good comments to improve our manuscript.

2- the inflammatory response has been evaluated using GFAP, which is an astrocyte marker. I think it would be more convenient to use a microglia marker such as IBA1 to assess the inflammatory process more efficiently and truthfully.

Reply: We agree with you. We also studied the expression of CD68 and IBA1 to confirm the activation of microglia; however, these proteins had low expression levels, so only GFAP with a relatively high expression level was used in this study. For future research, we plan to report the expression of microglia by implementing more diverse experimental conditions.

3-dental breakage used in the methodology (TEX) implies a greater risk of oral infections for animals. Have the authors excluded this possibility by evaluating any marker of infection?

Reply: As shown in the research method, TEX was performed twice a week during the course of the experiment, and during this process, the tooth was cut using sterilised fine scissors under anesthesia to prevent damage to the oral cavity. Although the oral cavity was observed closely during the operation, no inflammatory lesions were observed.

4-Only 9 references of the 31 used are after 2018. I think the authors should carry out a deeper and more updated bibliographic review.

Reply: In the revision process, several sentences were modified in the Discussion, and as per your comments, relatively recently published papers were added as references.

5-The article presents a large number of results of valuable interest, however the discussion of these results is superficial, and requires further work and deepening in which all the results obtained are framed.

Reply: As suggested, many sentences in the manuscript have been modified. Thank you very much for your comments, which improved the quality of our manuscript.

Reviewer 3 Report

In the article entitled "Tooth-cutting-induced Maxillary Malocclusion Exacerbates Cognitive Deficit in Mouse Model of Vascular Dementia," the authors investigate the effects of tooth cutting on cognitive function in mice undergoing carotid artery stenosis.

Several points are important to address.

1. There was a lack of a TEX-only control. That is, the mouse did not undergo carotid stenosis.

2.         In the results in Figure 4, a comparison between BCAS and BCAS+TEX would have been interesting.

3.         In Figure 6, why are no changes in gene expression of pro-inflammatory proteins visualized?

The study has several limitations, such as investigating the potential causes of cognitive deficit. Long-term effects of TEX were also not investigated.

It is important to determine how much TEX alone causes a cognitive and neuroinflammatory deterioration process.

In my view there is insufficient evidence of TEX on exacerbation of cognitive deficit.

The information would correspond to perhaps a pilot study.

Author Response

In the article entitled "Tooth-cutting-induced Maxillary Malocclusion Exacerbates Cognitive Deficit in Mouse Model of Vascular Dementia," the authors investigate the effects of tooth cutting on cognitive function in mice undergoing carotid artery stenosis.

Several points are important to address.

  1. There was a lack of a TEX-only control. That is, the mouse did not undergo carotid stenosis.

Reply: As per your opinion, we (the authors) conducted an experiment by separately classifying the experimental group that only received TEX treatment; as a result, no significant difference was found when compared with the sham operation group, so it was excluded from the manuscript. However, following authors’ discussion of your comments, it would be better to also include those results in the revised manuscript. Accordingly, we presented them as a supplementary material. Thank you for your good comments to improve our manuscript.

  1. In the results in Figure 4, a comparison between BCAS and BCAS+TEX would have been interesting.

Reply: Thank you very much for your comments to improve the quality of our manuscript. The result of our statistical analysis revealed no significant difference between the above two experimental groups; thus, these details were not separately described in the manuscript. However, following authors’ discussion of the raised point, these results were added in the revised manuscript to present the research results more clearly; thus, some parts of the experimental results were modified.

  1. In Figure 6, why are no changes in gene expression of pro-inflammatory proteins visualized?

Reply: In this study, the overall analysis of the genes whose expression was changed by BCAS and TEX was limited. However, as you pointed out, in our follow-up study, we would like to report the results of Western blotting and polymerase chain reaction on key proteins and genes involved in the inflammatory response.

The study has several limitations, such as investigating the potential causes of cognitive deficit. Long-term effects of TEX were also not investigated.

It is important to determine how much TEX alone causes a cognitive and neuroinflammatory deterioration process.

In my view there is insufficient evidence of TEX on exacerbation of cognitive deficit.

The information would correspond to perhaps a pilot study.

Reply: We agree with you on the study limitations raised. After discussing how TMJ imbalance can affect the aging process of older people for a relatively long period, we recently developed a method that can be applied to rodent models of ischemic stroke or vascular dementia. So far, it is true that our research results are insufficient; however, these limitations can be gradually solved through follow-up studies. Thank you very much for your constructive comments.

Round 2

Reviewer 2 Report

The authors response and modifications are satisfactory.

Reviewer 3 Report

Although I still think it is a pilot study, the authors did important changes in the article that make it more clear.